# Role of T Cells in Vaccine-Mediated Immunity against Marek’s Disease

**DOI:** 10.3390/v15030648

**Published:** 2023-02-28

**Authors:** Mohammad Heidari, Huanmin Zhang, Lakshmi T Sunkara, Syed Mudasir Ahmad

**Affiliations:** 1Avian Disease and Oncology Laboratory, Agriculture Research Service, Department of Agriculture, East Lansing, MI 48823, USA; 2Clemson Veterinary Diagnostic Center, Clemson University, Columbia, SC 29229, USA; 3Division of Animal Biotechnology, Faculty of Veterinary Sciences and Animal Husbandry, Srinagar 190006, India

**Keywords:** Marek’s disease, Marek’s disease virus, T lymphocytes, vaccination, T cell depletion

## Abstract

Marek’s disease virus (MDV), a highly cell-associated oncogenic α-herpesvirus, is the etiological agent of T cell lymphomas and neuropathic disease in chickens known as Marek’s disease (MD). Clinical signs of MD include neurological disorders, immunosuppression, and lymphoproliferative lymphomas in viscera, peripheral nerves, and skin. Although vaccination has greatly reduced the economic losses from MD, the molecular mechanism of vaccine-induced protection is largely unknown. To shed light on the possible role of T cells in immunity induced by vaccination, we vaccinated birds after the depletion of circulating T cells through the IP/IV injection of anti-chicken CD4 and CD8 monoclonal antibodies, and challenged them post-vaccination after the recovery of T cell populations post-treatment. There were no clinical signs or tumor development in vaccinated/challenged birds with depleted CD4^+^ or CD8^+^ T cells. The vaccinated birds with a combined depletion of CD4^+^ and CD8^+^ T cells, however, were severely emaciated, with atrophied spleens and bursas. These birds were also tumor-free at termination, with no virus particles detected in the collected tissues. Our data indicated that CD4^+^ and CD8^+^ T lymphocytes did not play a critical role in vaccine-mediated protection against MDV-induced tumor development.

## 1. Introduction

Marek’s disease (MD) is a highly contagious lymphoproliferative disease of domestic chickens caused by a cell-associated α-herpesvirus, Marek’s disease virus (MDV) [1]. Clinical signs of MD include depression, weight loss, transient paralysis, immunosuppression, and lymphoma formation in the visceral organs [2]. The early pathogenesis of MDV includes a lytic infection in B cells, followed by a latent infection in activated CD4^+^ T cells. This phase of infection could last up to three weeks before the reactivation phase of MDV infection, which is characterized by the second wave of lytic infection, permanent immunosuppression, and the transformation of T cells [2,3,4,5]. MDV strains have been categorized into three species based on biological features and genomic differences. The oncogenic strains and their attenuated forms are classified as Gallid alphaherpesvirus 2 (GaAHV2). The non-oncogenic strains isolated from chickens (SB1) and turkeys (HVT) are classified as Gallid alphaherpesvirus 3 (GaAHV3) and Meleagrid alphaherpesvirus 1 (MeAHV1), respectively [6,7]. The attenuated GaAHV2, CVI988/Rispens and non-oncogenic GaAHV3 and MeAHV1 have been used as vaccines to control MD since the 1960s [8,9,10,11]. CVI988/Rispens induces high protection efficacy and strong immunological responses against MDV infection in comparison to GaAHV3 and MeAHV1. This is speculated to be due to the genetic similarity of CVI988 to the oncogenic strains of MDV [12]. A shared intriguing feature of MD vaccine strains is that they all prevent lymphoma formation and induce a significant reduction in virus assembly and replication in the feather follicle epithelial cells (FFE), but they do not induce sterile immunity and consequently, virus particles remain in the host and are disseminated into the environment via stratified squamous epithelium or molted feathers [13,14,15,16,17].

Despite the significant success of vaccination in controlling MD, the molecular mechanism of vaccine-induced immunity is not well known and the potential role of the innate or adaptive immunity is unexplored. It is, however, speculated that shortly after vaccination, the activated natural killer (NK) cells induce apoptosis of infected B cells and the production of IFN-γ, which in turn activates macrophages, leading to the production of nitric oxide (NO) that has direct inhibitory effects on viral replication and infection [18]. It is believed that the role of adaptive immunity in protection mediated by vaccination is minimal [19]. NK cells are the cellular components of the innate immune system that effectively recognize and destroy tumor cells and cells infected with intracellular pathogens [20,21,22]. Although the biological functions of avian NK cells have not been fully characterized and their cellular markers are not completely identified, studies have shown a consistent enhancement and increased level of NK cell cytotoxicity in MD-resistant chickens in comparison to susceptible birds. Increases in NK cells’ activities have also been reported in both vaccinated MD-resistant and susceptible chickens that are associated with better vaccine-mediated protection [23,24,25]. It is believed that genetic resistance to MD is due to the superior activities of NK cells in the resistant birds [26]. Suppression of NK cells in the MD-susceptible line and elevated reactivation in the resistant line is suggestive of an essential role for NK cells in resistance to MD [25]. A recent study by Bertzbach et al. has revealed for the first time that MDV infects chicken NK cells, which leads to the activation, degranulation, and IFNγ production by these key players of the innate immune system [27]. IFNγ exerts an indirect antiviral action through the activation of macrophages leading to the NO production [28,29]. The inhibitory effect of NO on viral replication is believed to be the major mechanism of macrophages in controlling herpes virus infection [29]. NO has also been shown to inhibit MDV replication with the highest level of its production in the serum and spleen of MD-resistant birds [30,31]. Additional studies have also shown that the inhibition of NO synthase in chickens leads to an increase of viral load, suggesting a critical role of NO in controlling MDV replication [30]. There is growing concern that due to the continuous emergence of highly pathogenic strains of MDV and the lack of sterile immunity induced by vaccination, the vaccines currently available will not provide sufficient protection against emerging MDV strains. [18,32,33,34,35,36]. Therefore, MD remains a major economic threat to the global poultry industry and consequently, an elucidation of the immune mechanisms of vaccine-induced protection and development of recombinant vaccines with superior immunogenic features is of critical importance.

To provide insight into the possible role of T cells in vaccine-induced immunity, we depleted circulating T cells using anti-chicken CD4 or CD8 specific monoclonal antibodies (mAbs) and vaccinated the birds with depleted CD4^+^, CD8^+^, or CD4^+^/CD8^+^ T cells, followed by challenging with a very virulent strain of MDV. Our data indicate that neither CD4^+^ T cells nor CD8^+^ T cells play an essential role in preventing the tumor development induced by MDV.

## 2. Material and Methods

### 2.1. Experimental Chickens

The specific-pathogen-free white leghorn chickens used in this study were from the highly inbred MD-susceptible F1 progeny of lines 15I_5_ males and 7_1_ females (15I_5_ × 7_1_) that are maintained at the Avian Disease and Oncology Laboratory (ADOL) poultry farm. The 15I_5_ × 7_1_ birds were from unvaccinated breeder hens and carried no maternal antibodies to MDV or turkey herpesvirus. Chicks were hatched at the ADOL poultry facility and housed in modified Horsfall-Bauer isolation units for the duration of the experiment. All animal experiments were approved and carried out in accordance with the guidelines set forth by the Avian Disease and Oncology Laboratory Institutional Animal Care and Use Committee and the Guidelines for Care and Use of Laboratory Animals published by the Institute for Laboratory Animal Research (ILAR Guide) in 1996 (http://academic.oup.com/ilarjournal/article/38/1/41/664018, accessed on 15 March 2021).

### 2.2. Viruses

A bacterial artificial chromosome (BAC)-cloned very virulent (vv) strain of MDV, rMd5, which is propagated and maintained at ADOL, was used as the challenge virus in this experiment [37]. The CVI988/Rispens vaccine was purchased from Intervet (USA). Chickens were inoculated intraperitoneally with 2000 plaque-forming units (PFU) of the vaccine virus at day 12 post-hatch per manufacturer’s instruction. The birds were challenged 17 days post-vaccination with 1000 PFU of rMd5.

### 2.3. Hybridoma Cell Cultures

Lc-6 and Lc-4 hybridoma cell cultures [38,39,40,41] secreting ani-chicken CD4 and CD8, respectively, were maintained in Dulbecco’s Modified Eagle’s Medium (DMEM), as previously described [42]. The hybridoma cells were initially maintained in DMEM with 20% fetal bovine serum (FBS) that was gradually reduced to 10%, 5%, and 2% (Thermo Fisher Scientific, Waltham, MA, USA) every three days. For antibody collection, cells were maintained in protein-free hybridoma medium II (PFHMII) with 0.2% chemically defined lipid concentrate (Life Technologies, Carlsbad, CA, USA). For large scale mAb production, the Wheaton CELLine 1000 bioreactor flask cells were used following the manufacturer’s protocols (Wheaton, IL, USA).

### 2.4. Purification of Monoclonal Antibodies

The purification and characterization of anti-chicken CD4 and CD8 mAbs were performed as described by Umthong et al. [42]. The supernatants from cell cultures of Lc-6 and Lc-4 were collected every 3–4 days and proteins were precipitated by saturated ammonium sulfate solution (Thermo Fisher Scientific). The supernatants were centrifuged, and the pellets were resuspended in 5–8 milliliters of PBS, depending on the size of the pellets. The protein suspensions were dialyzed against PBS overnight at 4 °C in dialysis cassettes (Slide-A-Lyzer, Thermo Fisher Scientific). The modified route of delivery and concentrations of mAbs needed for the efficient depletion of CD4^+^ and CD8^+^ T cells were based on the study by Umthong et al. [42].

### 2.5. Peripheral Blood Mononuclear Cells Isolation

At 11 and 28 days of age, three chickens from each group were bled via jugular vein (0.5–1 mL). Due to the small size of the young birds, the anticoagulated blood samples from each group were pooled and 1.5–3 mL of each sample was mixed with an equal volume of phosphate buffer saline solution (PBS) and layered onto 3–6 mL of Histopaque-1077 (Sigma-Aldrich, St. Louis, MO, USA) in a 15 mL conical centrifuge tube and centrifuged at 400× *g* for 30 min at room temperature. Peripheral blood mononuclear cells (PBMC) were aspirated from the interphase, diluted with 10 mL of isotonic PBS, and pelleted by centrifugation at 250× *g* for 10 min. The PBMC were washed three times in PBS by resuspension of pellet and centrifugation at 250× *g* for 10 min each.

### 2.6. Monoclonal Antibodies Used in Flowcytometric Analysis of PBMC

The monoclonal antibody used for the detection of chicken B cells (mouse anti-chicken Bu1-RPE) and CD4^+^ T cells (mouse anti-chicken CD4-PE) were purchased from Southern Biotech (Birmingham, AL, USA). The monoclonal antibody for the detection of CD8α^+^ T cells (CD8α-FITC, 11–39) was purchased from ThermoFisher Scientific. The secondary monoclonal antibodies used in the binding specificity of anti-CD4 mAb isolated from hybridoma cell line, were purchased from Southern Biotech (rat anti-mouse IgM-PE/CY7).

### 2.7. Flowcytometric Analysis of PBMC

Subpopulations of the isolated PBMC from pooled blood samples were quantified based on the expression pattern of cell surface antigens. Aliquots of 1 × 10^6^ PBMC in 100 μL of FACS buffer were added to a 96-well plate and incubated with specific monoclonal antibodies for 30 min at 4 °C. Cells were washed three times with 200 μL of FACS buffer. The washed cells were resuspended in 200 μL of FACS buffer and analyzed by flowcytometry. A FACScan flowcytometer from Becton Dickinson (Mountainview, CA, USA) was used for the cell surface analysis.

### 2.8. Characterization of Anti-CD4 and Anti-CD8 Monoclonal Antibodies

The characterization of anti-CD4 and anti-CD8 mAbs has previously been reported [38,39,40,43,44]. Further characterization of the isolated anti-CD8 mAb was performed by Umthong et al. [42]. The additional binding specificity of anti-CD4 mAb (IgM isotype) isolated from the supernatant of hybridoma cell line (Lc-6) was evaluated by flow cytometric analysis in comparison to commercially available anti-CD4 mAb (Southern Biotech, Birmingham, AL, USA). PBMC were purified from a whole blood sample of one adult chicken using Histoplaque-1077 (Sigma Aldrich, St. Louis, MO, USA). The cells were stained with the primary mAb isolated from hybridoma cell line at 1/20 and 1/100 dilutions for 30 min at 4 °C. Cells were washed and then counter stained with the secondary mAb (Rat anti-mouse IgM, PE/CY7, Southern Biotech) for 30 min at 4 °C. Cells were washed twice and resuspended in 200 μL of FACS buffer for flow cytometric analysis.

### 2.9. PCR Analysis of MDV-Encoded pp38 Gene

The DNA samples isolated from spleen tissues were used for PCR analysis of the MDV-encoded pp38 gene. The pp38 primers are specific for pathogenic strains of MDV and will not bind to CVI988/Rispens DNA. One μL of DNA suspension at 50 ng/μL was used in a 25-μL reaction in the presence of 10 pmol of each primer and 12.5 μL of 2× PCR mix (Promega, Madison, WI, USA). The amplification of pp38 and the housekeeping gene, GAPDH, was performed with an initial denaturation at 94 °C for 2 min followed by 30 cycles of denaturation (94 °C for 30 s), annealing (58 °C for 30 s), extension (72 °C for 30 s), and a final single cycle of extension for 10 min at 72 °C. The expression of the housekeeping gene, GAPDH, was used as a control. The primers used for GAPDH and pp38 amplification are as follows:

GAPDH F: 5′-ATG GGC ACG CCA TCA CTA TC-3′

GAPDH R: 5′-ACC CCA GCC TTC TC-3′

PP38 F: AAG GGT GAT GGG AAG GCG ATA G-3′

PP38 R: GCA TAC CGA CTT TCG TCA AGA TG-3′

### 2.10. MDV Genome Copy Number Assay

MDV genome copy number analysis was performed according to the previously described protocol [45,46]. Analysis of genomic DNA from each skin sample was performed in triplicate using quantitative Real-Time PCR (qPCR). Primers for both chicken GAPDH and MDV-encoded gB were each used at 0.5 µM and the probes for MDV gB and chicken genes were used at 0.2 µM. MDV loads are depicted as the copy number of MDV gB divided by the copy number of chickens’ GAPDH. A Dunnett’s multiple comparison test (one-way ANOVA) was run between the control group (unvaccinated and unchallenged animals) and every other trial group at 57 dpi (termination) using GraphPad PRISM software (GraphPad, La Jolla, CA, USA).

### 2.11. Immunohistochemistry

Samples previously flash frozen in embedding medium, Optimal Cutting Temperature (OCT) (Sakura Finetek, Torrance, CA, USA), were sectioned on a cryotome at 5 mm and placed on slides coated with 2% 3-Aminopropyltriethoxysilane and air dried at 25 °C overnight. Subsequently, microtome sections were fixed in formal acetate fixative for 10 min at room temperature followed by three changes of Tris buffered saline—5 min each. Endogenous peroxidase activity was blocked with 0.3% Hydrogen peroxide in Tris buffered saline for 20 min followed by tap and distilled water rinses. Following pre-treatment standard, Avidin-Biotin complex staining steps were performed at room temperature on the DAKO Autostainer (Agilent Technologies, Carpentaria, CA, USA). All staining steps were followed by rinses in Tris buffered saline + Tween 20 (Scytek Laboratories, West Logan, UT, USA). After blocking non-specific proteins with normal horse serum (1/30 dilution in PBS; Vector Labs, Burlingame, CA, USA) for 30 min, sections were incubated with Avidin/Biotin blocking system for 15 min each (Vector Lab, Burlingame, CA, USA; Sigma, St. Louis, MO, USA). Samples were then incubated with mouse anti MDV gB monoclonal antibody [6] for 1 h in Normal Antibody Diluent (NAD) (Scytek Laboratories, West Log, UT, USA), followed by rinsing and incubation with biotinylated horse anti-mouse IgG (H + L), prepared at 11.5 lg/mL in NAD incubated for 60 min. Samples then were incubated with R.T.U. Vector Elite Peroxidase Reagent (Vector Laboratories, Burlingame, CA, USA) for 30 min. Reaction development utilized Vector Nova Red peroxidase chromogen incubation for 15 min followed by counterstain in Gill Hematoxylin (Thermo Fisher) for 15 s, differentiation, dehydration, and clearing and mounting with synthetic mounting medium. The working solution for the monoclonal antibody specific for MDV gB was 1:1000.

### 2.12. Statistical Analysis

Due to the small size of the birds, the blood samples isolated from three individual chickens were pooled and consequently, no statistical analysis could be performed. Therefore, the bar graphs represent relative changes in B and T cell populations. The viral genome copy number in the skin of infected birds, however, was based on comparative analysis between individual infected and control birds at 57 dpi. Statistical analysis was performed with GraphPad PRISM software (GraphPad, La Jolla, CA, USA) using Dunnett’s multiple comparison test (one-way ANOVA).

### 2.13. Experimental Design

One hundred and fifty-six one-day-old chicks from line 15I_5_ × 7_1_ were randomly distributed into six groups of 26 birds each in separate isolators (Table 1, A–F). The birds in group A were untreated, unchallenged, and served as the negative control. The birds in group B were untreated with intact T cells, vaccinated, and challenged along with the treated birds. The birds in group F were untreated, unvaccinated, but challenged and served as the positive control for MDV-induced infection. The birds in groups C, D, and E were treated with specific mAbs per Table 2. Briefly, the anti-CD4 and anti-CD8 antibody treatment started on day of hatch by intraperitoneal (IP), intravenous (IV), or combination of IP/IV routes with repeat injections every three days. The treatment continued for up to three days post-vaccination. Then the treatment was terminated to allow T cell recovery as target cells for the challenge virus. Flowcytometric analysis of PBMC from three birds from group A (control as baseline), and treated groups C, D, and E was performed to verify T cell depletion before vaccination and T cell recovery before challenge. The birds were observed for clinical sign of MD twice per week until the termination of the experiment at 57 days post challenge. At 5-, 10-, and 20-days post-infection (dpi), three chickens from each group were randomly selected and euthanized by CO_2_ inhalation for necropsy and tissue sample collection. At 57 dpi, the remaining chickens from each group were euthanized and necropsied for biopsy examination, tumor development assessment, and tissue collection. Spleen tissue samples were stored in RNAlater (Thermo Fisher Scientific) at −20 °C until used for DNA analysis and the detection of viral genome. Skin samples were flash frozen in embedding medium and stored at −80 °C until used for immunohistochemical analysis. Approximately 10 mg of spleen tissues were used for DNA isolation using Qiagen DNeasy Blood and Tissue Kit according to the manufacture’s recommendation. Skin DNA isolation was performed according to previously published protocol [47].

## 3. Results

### 3.1. Purification and Binding Specificity of Anti-CD4 and Anti-CD-8 mAbs

The mAbs against chicken CD4 and CD8 were produced by culturing Lc-4 (anti-CD8) and Lc-6 (anti-CD4) hybridoma cell lines in Celline 1000 bioreactor flasks. The binding specificity of anti-CD8 mAb was confirmed by Umthong et al. [42]. Although the detailed characterization of both mAbs have previously been reported, further binding specificity of anti-CD4 mAb was confirmed by flow cytometric analysis using PMNC.

### 3.2. T Cell Depletion

To confirm that in vivo antibody treatment of chicks was effective in the considerable reduction of circulating CD4^+^ and CD8^+^ T cells, flow cytometric analysis of pooled PBMC from three birds per control (baseline) and treated groups was performed 11 days post the first antibody treatment on the day of hatch. Additionally, to verify the recovery of T cells post-termination of antibody treatment, flow cytometric analysis was once more performed on PBMC cells of three birds from the above-mentioned groups 13 days after the last round of antibody treatment. The effect of T cell depletion on B cell population was also assessed.

### 3.3. Flowcytometric Analysis of PBMC at 11 Days Post Antibody Treatment

To verify the effectiveness of antibody treatment in the complete or substantial depletion of circulating T cells, we analyzed the CD4^+^ and CD8^+^ T cell populations in the pooled blood samples of three control and three treated birds after 11 days of antibody treatment. Data from the flow cytometric analysis shows that the population of CD4^+^ T cells in the anti-CD4 antibody treated birds was reduced to 0.019% of the total lymphocytes in the tested blood sample compared to the untreated control birds at 5.06% (Figure 1A,B). Although the depletion of CD8^+^ T cells was not as effective as that of CD4^+^ T cells, there was a large reduction in the population of CD8^+^ T cells in the anti-CD8 antibody treated birds (Figure 1C). A similar result was obtained in the combined CD4^+^/CD8^+^ T cell depletion. The CD4^+^ T cells were completely depleted, while the CD8^+^ population was at 1.63% of the total lymphocytes in the tested blood samples (Figure 1D). The bar graphs in Figure 1E depict the reduction of T cells at 11 days post-antibody treatment. The effect of T cell depletion on the population of B cell was also assessed.

### 3.4. Flowcytometric Analysis of PBMC at 13 Days Post Termination of Antibody Treatment

To provide target cells for MDV, the antibody treatment for depletion of T cells was terminated three days post vaccination. Considering the lingering effect of residual antibodies in neutralizing T cells in the treated birds, we allowed 13 days for the recovery of T cells post treatment before challenging the birds. Flow cytometric data indicates that the CD4^+^ T cells population in the anti-CD4 antibody treated birds increased to 4.52% compared to the control birds at 8.72% (Figure 2). The CD8^+^ T cell population in these birds, however, was more than twice as those of the control birds (10.6% vs 4.78%). In contrast, the CD8^+^ T cell population in the anti-CD8 antibody treated birds increased to 6.5%, slightly over those of the control birds. The CD4^+^ T cells in these latter birds, however, rose to the level of those in the control birds. The bar graphs in Figure 2D show the recovery of T cells 13 days post termination of antibody treatment. The effect of T cell depletion and recovery on the population of B cell was also assessed.

### 3.5. Detection of Viral Genome in the Spleen Tissues at 5-, 10-, 20-, and 57-dpi

PCR analysis of spleen tissues from control, treated, and nontreated vaccinated challenged birds was performed to detect viral genome presence. Figure 3 shows the PCR results at 5-, 10-, 20-, and 57 dpi (Panels A, B, C, and D, respectively). The primers for MDV-encoded pp38 (128 bp) gene were used for analysis. GAPDH (203 bp) was used as a housekeeping control gene. The viral genome detection in the non-vaccinated challenged birds (Lanes 14, 15, and 16) is depicted by green arrows. The detection of pp38 in the T cell depleted, vaccinated, and challenged birds (Lanes 2–13) is shown by red arrows. 

### 3.6. Flow Cytometric Analysis of Binding Specificity of Anti-CD4 mAb

Figure 4 depicts the specific binding of purified anti-CD4 mAb (Lc-6) isolated from hybridoma cell line. Anti-CD4-PE mAb from Southern Biotech was used as a positive control (panel B). At two different dilutions of 1/20 and 1/100, the Lc-6 mAb specifically stained the same CD4^+^ T cell population stained with mAb from Southern Biotech (panels D and E). Panels A and C are negative controls, with no mAb and secondary mAb only, respectively.

### 3.7. MDV Genomes Copy Number in the Skin Tissues

DNA samples isolated from the skin tissues of three birds from each group at 57 dpi were used for viral genome copy number analysis. Real-time PCR analysis revealed the highest virus load in the skin tissues of non-vaccinated challenged birds confirming the result of immunohistochemical analysis (Table 3 and Figure 5). The CD4^+^ T cell depleted vaccinated/challenged birds showed a much lower viral load in comparison to the non-vaccinated/challenged birds. No virus genome was detected in either CD8^+^ or CD4^+^/CD8^+^ T cell depleted vaccinated/challenged birds. Despite a minor replication of the virus in the skin of these birds, the PCR analysis of viral DNA at 57 dpi was negative for all vaccinated birds with depleted or intact T cells. This discrepancy is likely due to the skin samples used for DNA isolation.

### 3.8. Immunohistochemical Analysis of MDV Replication

To detect viral antigen in the skin tissues of vaccinated/challenged or challenged only groups, anti-gB (MDV-encoded glycoprotein) monoclonal antibody was used. Immunohistochemical analysis shows a massive replication of MDV in the FFE of the unvaccinated, untreated group at termination (Figure 6, panel A, blue arrows). The vaccinated/challenged birds with intact T cells exhibited far less replication of virus particles in the FFE than those in the non-vaccinated/challenged group (Figure 6, panel B, blue arrows). The CD4^+^ T cell depleted vaccinated/challenged birds also had very few virus particles replicating in the FFE (Figure 6, panel C, blue arrow). The CD8^+^ T cell depleted vaccinated/challenged birds showed slightly more virus replication than the CD4^+^ T cell depleted birds (Figure 6, panel D, blue arrows). Although the FFE of the CD4^+^/CD8^+^ T cell depleted vaccinated/challenged birds exhibited some non-specific binding (green arrows), it is safe to conclude that the level of virus particle replicating in the FFE is roughly at the same level as the CD8^+^ T cell depleted birds (Figure 6, panel E, blue arrow). 

### 3.9. Protection Efficacy of CVI/988 Rispens in Birds with Depleted T Cells

The vaccine-induced protection in the birds with depleted CD4^+^, CD8^+^, and CD4^+^/CD8^+^ T cells at termination is shown in Table 4. As expected, the vaccinated challenged birds with intact T cell population were protected with no clinical sign of MD or MDV-induced tumor development at 57 dpi (second row). The non-vaccinated/challenged birds with intact T cells developed MD with tumors in their visceral organs (last row). The CD4^+^ and CD8^+^ T cell depleted vaccinated/challenged birds were protected with no clinical signs or tumor development at termination (Rows 3 and 4, respectively). The CD4^+^/CD8^+^ T cell depleted, vaccinated, and challenged birds, however, were depressed, extremely emaciated, and exhibited difficulty in breathing (Figure 7). Necropsy at termination, however, did not show any sign of MDV-induced tumors, and PCR analysis of spleen tissues was negative for the presence of MDV genome (Figure 3 and Figure 5). Unlike MDV-induced splenomegaly (enlargement of the spleen), the spleen tissues in these birds were severely atrophied (Figure 7B). The bursa of Fabricus, however, was atrophied like in MDV-infected birds (Figure 7C). It is not clear at this point whether the unusual pathological features observed in these birds were due to T cell depletion or associated with a secondary infection. 

## 4. Discussion

MDV, the etiological agent of MD, is a cell-associated α-herpesvirus that causes paralysis, anemia, immunosuppression, and T cell lymphomas in visceral organs of domestic chickens [13]. Although vaccines have been used to control MD since the late 1960s, the molecular mechanism of vaccine-induced protection is not well known. Vaccination prevents MDV-induced tumor formation, but not infection and transmission [48,49]. The continuous evolution of MDV, however, has been a major concern for the poultry industry in that the existing vaccines might not be effective enough to control the newly evolved and highly pathogenic strains of MDV [32,50]. Deciphering the underlying mechanism of immunity induced by vaccination would be essential in developing more efficacious vaccines that could provide sterile immunity and prevent virus dissemination and future outbreaks.

In contrast with most animal and human vaccines that require the activation and direct involvement of the adaptive immune system for the induction of adequate protection, MD vaccines provide partial immunity within the first 24–48 h post-vaccination [51]. With high protective efficacy, a vaccine could induce 95–100% protection by day five post-vaccination [51]. Unlike the adaptive immune system that might take 7–10 days for the activation and induction of B and T cells-mediated immunity, the innate immune responses are rapidly induced and play a critical role in the control of viral infection and replication [52]. Among the cellular components of the innate immune system, the NK cells are believed to be the likely candidate cells involved in vaccine-mediated protection [53,54].

Our previous study has revealed that B cells do not play an essential role in vaccine-induced protection [47].The goal of the present study was to provide insight into the role of T cells in immunity mediated by vaccination. Using specific monoclonal antibodies, birds were depleted of CD4^+^, CD8^+^, or combined CD4^+^/CD8^+^ T cells followed by vaccination. The antibody treatment that started on the day of hatch was terminated three days post-vaccination to allow T cell recovery as the target cells for the MDV challenge that followed 13 days post-treatment termination (Table 1 and Table 2).

Although there was a substantial reduction in the population of circulating CD8^+^ T cells, the antibody-mediated depletion of these cytotoxic T cells was not as effective as that of the CD4^+^ T cells (Figure 1). One possible explanation for the observation is the involvement of CD4^+^ T cells in B cell activation and isotype switching. The CD4^+^ T cells recognize the non-polysaccharide antigens presented by B cells via MHC class II that leads to the generation of high affinity antibodies for T cell-dependent antigens [55,56]. In our study, however, the anti-CD4 mAb was so effective in neutralizing the CD4^+^ T cells that there were no functional CD4^+^ T cells left to interact with B cells to result in the generation of antibodies counteracting the anti-CD4 mAb. Therefore, the anti-CD4 mAb was effective in the near complete depletion of CD4^+^ T cells. In the case of CD8^+^ T cell depletion, however, the CD4^+^ T cells were free to interact and induce activation and isotype switching in B cells, resulting in the generation of antibodies against anti-CD8 mAb (protein) and neutralizing its functional activity in the depletion of CD8^+^ T cells.

As the data indicate, the vaccination in the presence of markedly reduced CD4^+^ and CD8^+^ T cells provided 100% protection against tumor development induced by a very virulent strain of MDV (Table 3). These birds, like the vaccinated/challenged birds with intact T cells, showed no clinical sign of MD during the experimental period or T cell lymphomas at termination. Although the birds with depleted combined CD4^+^/CD8^+^ T cells were also free of clinical signs of MD or tumor development, they were severely emaciated and exhibited difficulties in breathing (Figure 7). It is likely that the severe reduction of combined CD4^+^/CD8^+^ T cells in these birds left them susceptible to respiratory or other types of infections. Morimura et al. [41,43] vaccinated thymectomized birds but did not observe such ailment in challenged birds following vaccination. The residual circulation of CD4^+^ or CD8^+^ T cells, however, was depleted separately via specific anti-CD4 or anti-CD8 mAbs, but not both cell types at the same time. An additional comparative study between infected and non-infected vaccinated CD4^+^/CD8^+^ T cell depleted birds will determine if the phenotype is due to combined T cell depletion or induced by a secondary bacterial or viral infection.

A recent study by Umthong et al. indicates that the deletion of CD8^+^ T cells in vaccinated and challenged chickens results in a higher incidence of tumor development [42]. This is a clear contradiction to our data and earlier published studies. This apparent discrepancy rises from the misinterpretation of the presented data. The reported data from Umthong et al. show a higher incidence of tumor development in CD8^+^ T cell depleted, HVT or SB-1 vaccinated and challenged birds. Yet, there are no sign of tumors in the CD8^+^ T cell depleted, bivalent (HVT+SB-1) vaccinated, and challenged birds. This is an obvious indication that a vaccine with high protection efficacy can provide immunity against tumor development in chickens with significantly reduced or depleted CD8^+^ T cells. The tumor development in the CD8^+^ T cell depleted, HVT or SB-1 vaccinated birds in the above-mentioned study is not due to the CD8^+^ T cells depletion, but the failure of the vaccine to induce protection. Previous studies have also shown that HVT or SB-1 provide only partial protection against MDV [1,57]. Additionally, investigation by Morimura et al. has provided evidence that CD4^+^ and CD8^+^ T cells do not play an essential role in anti-tumor immunity induced by vaccination [41,43]. The depletion of T cells by neonatal thymectomy, however, has been shown to reduce the incidence of MDV-induced tumors in challenged birds, indicating that T cells serve as target cells for MDV [58]. On the other hand, a study by Calnek et al. has shown that suppressive treatments with neonatal thymectomy and cyclophosphamide administration did not influence transplanted tumor growth and rejection in MD susceptible chickens, indicating a functional role for T cells in the rejection of transplants [59]. Although our data indicate that T cells do not play an important role in the immunity against tumor development provided by vaccination, the CD8^+^ T cells along with NK cells play critical roles in immune defense against intracellular pathogens [60]. The production of toxic mediators including perforin and granzymes facilitate the apoptosis of target cells and, therefore, prevent the survival of invading pathogens [60]. Furthermore, it is believed that NK cells, γδ T cells, and NK-like T cells contribute to the protection mechanism mediated by vaccination [40,41,44]. The adoptive transfer of TCRγδ-activated PBMC was shown to reduce virus replication and MDV-induced tumor development [61]. Another study by Hao et al. provides evidence that the vaccination of chickens with CVI988 induces a significant expansion of γδ and CD8α^+^ T cells in different tissues post-vaccination. It was also shown that the expansion of these cells was CVI988-specific as the pathogenic strain of MDV did not result in the proliferation of either cell type, suggesting a potential role of γδ T cells in vaccine-mediated immunity [62]. The data from the study, however, show that the secondary vaccination only induced memory CD8^+^ T cells but not γδ T cells. As for the potential role of NK cells in MD, it has been shown that the primary NK cells from chicken embryonic spleen were activated when co-cultured with MDV-infected chicken embryo cells, expressing CD107a, an activation marker for cytotoxicity and IFN-γ production [27]. Vaccination in the absence of NK cells and/or γδ T cells will provide evidence for the possible role of these cells in protection induced by immunization. Progress in transgenic techniques has enabled scientists to decipher the underlying mechanism of viral pathogenesis. The development of transgenic chickens lacking peripheral T cell populations or T cell receptors, as it has been done with B-cell knockout chickens [63,64], will serve as a unique experimental model to study the specific role of the T cell-mediated immune responses to avian infectious diseases.

Immunohistochemical analysis revealed that a massive number of virus particles were produced in the FFE of the non-vaccinated challenged birds (Figure 6A, blue arrows). The vaccinated challenged birds with intact T cells, however, had a considerably reduced number of virions produced in the FFE (Figure 6B, blue arrows). A similar pattern of viral replication was observed in the FFE of birds with depleted CD4^+^, CD8^+^, and CD4^+^/CD8^+^ T cells (Figure 6C, Figure 6D and Figure 6E, respectively; blue arrows). The staining within feather pulps is likely non-specific and does not represent viral antigens (Figure 6, green arrows).

PCR analysis, using primers from MDV-encoded pp38, was conducted to detect the presence of MDV DNA in the spleen tissues of treated and untreated vaccinated challenged birds at 5-, 10-, 20-, and 57-dpi. At 5 dpi, one of the three tested birds with depleted CD4^+^ T cells showed a weak band, representing challenge virus DNA (Figure 3, panel A, lane 5, red arrow). Viral DNA was also detected in the spleen tissues of all three tested birds with CD4^+^/CD8^+^ T cells (Figure 3, panel A, lanes 11–13, red arrows). All three non-vaccinated/challenged birds with intact T cells were also positive for the presence of viral DNA in their spleen tissues (Figure 3, panel A, lanes 14–16, green arrows). At 10 dpi, the spleen tissues of one of the three CD4 ^+^ T cells birds along with two of the CD4^+^/CD8^+^ T cell depleted birds were positive for viral DNA (Figure 3, panel B, lanes 3 and 11–12, respectively; red arrows). A similar pattern was observed at 20 dpi (Figure 3, panel C, lanes 6, and 11–13; red arrows). At termination, however, none of the treated and non-treated vaccinated challenged birds were positive for viral DNA (Figure 3, panel D, lanes 2–13).

The viral genome copy number in the skin of treated and non-treated vaccinated/challenged birds at 57 dpi (termination) supported the PCR and immunohistochemistry data (Table 3 and Figure 5).

In summary, it appears that T cells do not play a critical role in vaccine-mediated immunity against MDV-induced tumor development. It is likely that cellular components of the innate immune system, including NK cells and γδ T cells, contribute to protection mediated by vaccination. 

## Figures and Tables

**Figure 1 viruses-15-00648-f001:**
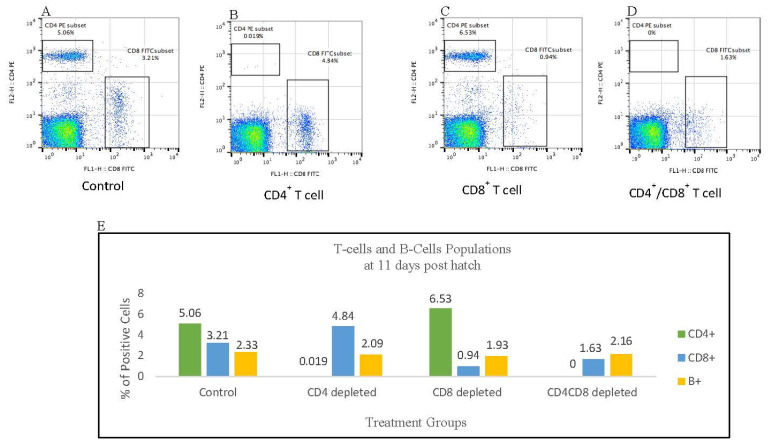
T cells depletion. Flow cytograms show the percentage of CD4^+^ and CD8^+^ T cells in the control (**Panel A**), CD4^+^ T cell depleted birds (**Panel B**), CD8^+^ T cell depleted birds (**Panel C**), and CD4^+^/CD8^+^ T cell depleted birds (**Panel D**) 11 days post-treatment. Blood samples from three birds per group were pooled, PBMC isolated, and 1 × 10^6^ cells/100 μL was used for cell surface antigen analysis. The CD4^+^ T cells were stained with CD4-PE, and CD8^+^ T cells were stained with CD8α-FITC, 11–39 monoclonal antibodies. (**Panel E**) Bar graphs showing the percentage of B and T cell populations 11 days after antibody treatment. Comparative analysis was made between the untreated control and the T cell depleted birds. Same total blood samples were used for the staining of B cells and double staining of CD4^+^ and CD8^+^ T cells. B cells, CD4^+^ T cells, and CD8^+^ T cells were stained with monoclonal antibodies Bu1-RPE, CD4-PE, and CD8α -FITC, respectively.

**Figure 2 viruses-15-00648-f002:**
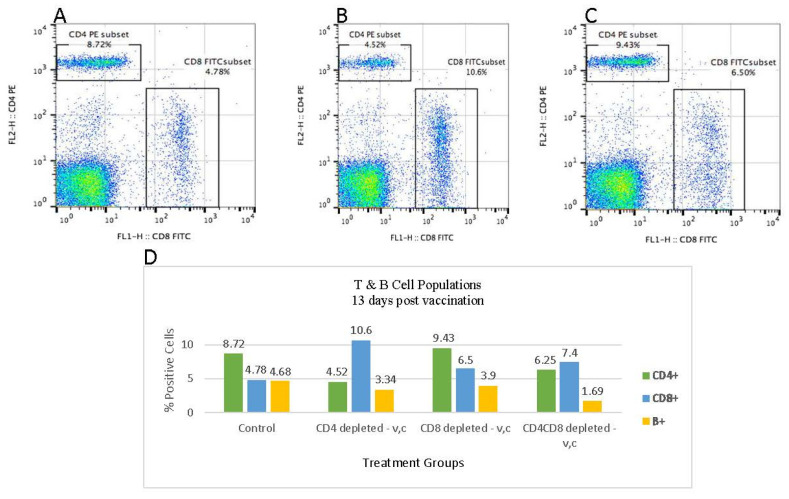
Recovery of CD4^+^ and CD8^+^ T cells 13 days post-termination of antibody treatment. The percentage population of CD4^+^ and CD8^+^ T cells in the control birds (**Panel A**), CD4^+^ T cell depleted birds (**Panel B**), and CD8^+^ T cell depleted group (**Panel C**) are depicted 13 days post-termination of antibody treatment. Blood samples from three birds per group were pooled, PBMC isolated, and 1 × 10^6^ cells/100 μL was used for cell surface antigen analysis. The CD4^+^ T cells were stained with CD4-PE, and CD8^+^ T cells were stained with CD8α -FITC, 11–39 monoclonal antibodies. (**Panel D**) Bar graphs showing the percentage of B and T cell populations 13 days after termination of antibody treatment. Comparative analysis was made between the untreated control and the T cell depleted birds. Same total blood samples were used for the staining of B cells and double staining of CD4^+^, and CD8^+^ T cells. B cells, CD4^+^ T cells, and CD8^+^ T cells were stained with monoclonal antibodies Bu1-RPE, CD4-PE, and CD8α -FITC, respectively. V: vaccinated; C: challenged.

**Figure 3 viruses-15-00648-f003:**
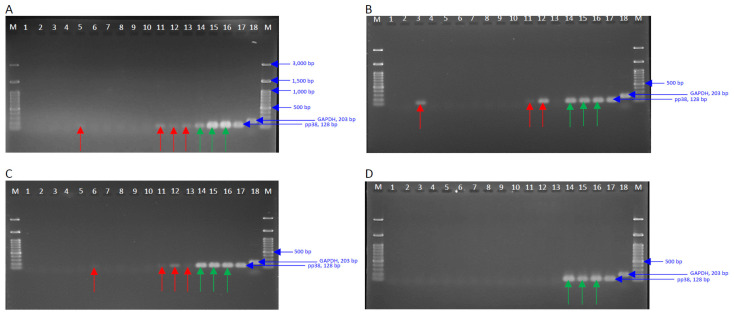
PCR-based analysis of viral DNA in spleen samples of control and treated birds at 5 days post-inoculation (dpi, **Panel A**), 10 dpi (**Panel B**), 20 dpi (**Panel C**), and 57 dpi (**Panel D**). The viral genome detection in the non-vaccinated challenged birds (Lanes 14, 15, and 16) is depicted by green arrows. The detection of pp38 in the T cell depleted, vaccinated, and challenged birds (lanes 2–13) is shown by red arrows. Lanes: M, DNA ladder, 1: Control bird, 2–4: Birds with intact T cell, vaccinated, challenged, 5–7: Birds with CD4+ T cell depleted, vaccinated, challenged, 8–10: Birds with CD8+ T cell depleted, vaccinated, challenged, 11–13: Birds with CD4+/CD8+ T cell depleted, vaccinated, challenged, 14–16: Birds with intact T cells, non-vaccinated, challenged, 17: Positive control for pp38 amplification using MDV DNA isolated from infected birds (blue arrow), 18: GAPDH (blue arrow), M: DNA ladder.

**Figure 4 viruses-15-00648-f004:**
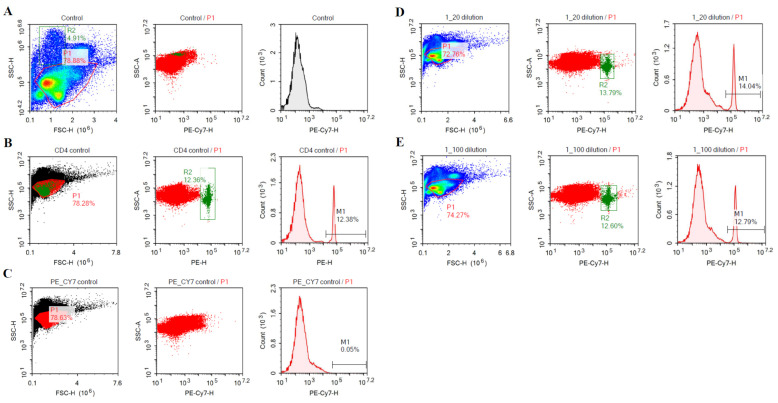
Anti-CD4 mononuclear cell binding specificity. (**A**): Histopaque 1077-treated PBMC (1 × 10^6^ cells) with no added antibodies (negative control); (**B**): PBMC stained with mouse anti-chicken CD4-PE antibody (Southern Biotech, positive control); (**C**): PBMC stained with rat anti-mouse IgM-PE/CY7 antibody (secondary antibody only); (**D**): PBMC stained with primary monoclonal antibody isolated from hybridoma cell line (IgM, at 1.5 µg per 1 × 10^6^ cells) and the secondary rat anti-mouse IgM-PE/CY7 antibody; (**E**): PBMC stained with primary monoclonal antibody isolated from hybridoma cell line (IgM, at 0.298 µg per 1 × 10^6^) and the secondary rat anti-mouse IgM-PE/CY7 antibody. The gated green cells in the middle of panels (**D**,**E**) are staining the same population of cells as in the middle of panel B.

**Figure 5 viruses-15-00648-f005:**
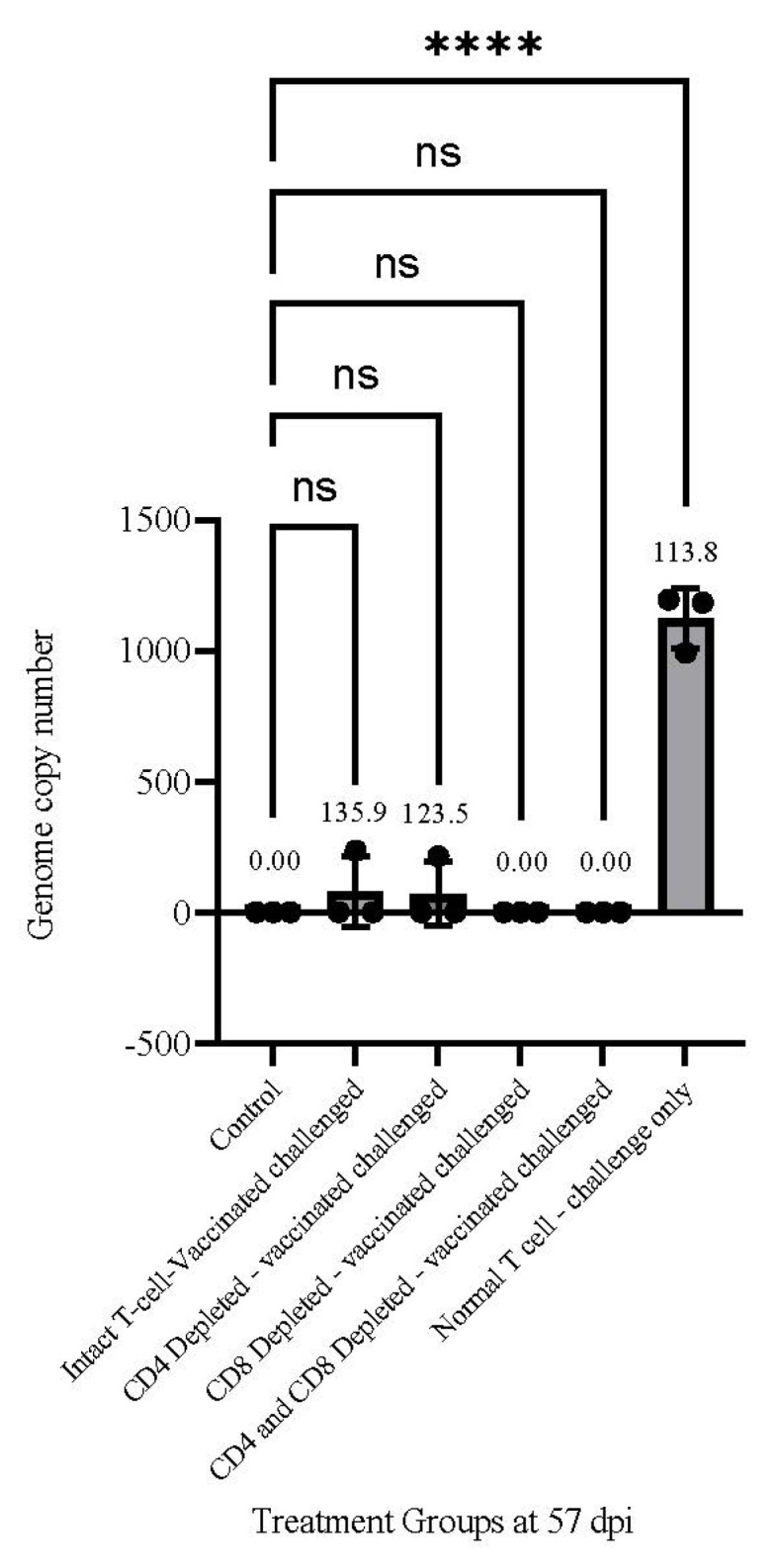
Viral genome copy number in the skin of infected birds at termination (57 dpi). DNA samples of the skin tissues from three individual birds from each group at 57 dpi were used for the determination of the viral genome copy number. The high genome copy number of MDV in the skin of the non-vaccinated challenged birds supports the result of immunohistochemical analysis. A Dunnett’s multiple comparison test (one-way ANOVA) was run between the control group (unvaccinated and unchallenged animals) and every other trial group at 57 dpi (termination). The normal T cell, challenge-only group showed a significant difference from the control group (**** *p* value of < 0.0001, ns: no significant).

**Figure 6 viruses-15-00648-f006:**
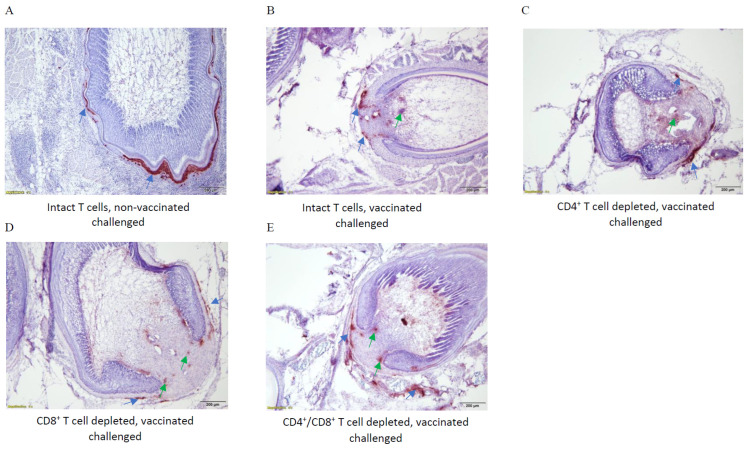
Immunohistochemical analysis of MDV antigen in the skin samples of all vaccinated and challenged groups with intact or depleted T cells. Anti-gB monoclonal antibody was used for detection of virus particles in the skin tissues of challenged groups. (**Panel A**) depicts skin sample from an unvaccinated, challenged bird with intact T cells showing significant viral replication in the FFE (blue arrow). (**Panel B**) represents the skin sample from a vaccinated/challenged bird with intact T cells showing minor MDV antigen in the FFE (arrows). (**Panel C**) depicts skin sample from a CD4^+^ T cell depleted, vaccinated/challenged bird that exhibits minor viral replication in the FFE (blue arrow). The replication rate of MDV in the skin of a CD8^+^ T cell depleted bird is depicted in (**Panel D**) (arrows). (**Panel E**) shows the replication rate of MDV in the skin sample of a CD4^+^/CD8^+^ T cell depleted, vaccinated/challenged bird (blue arrow). The staining within the pulp of feather shafts (**Panels B**–**E**) are non-specific staining and are not representing virus antigen (green arrows). Scale bar = 200 µm (**A**–**E**).

**Figure 7 viruses-15-00648-f007:**
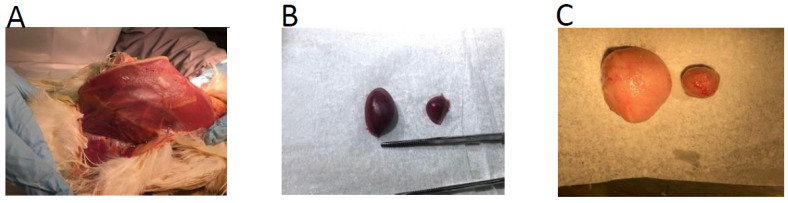
The picture depicts the chest bone (keeled sternum) of a CD4^+^/CD8^+^ T cell depleted bird that is severely emaciated (**Panel A**). These birds exhibit no clinical signs of MD during the experiment and no T cell lymphoma at termination. The birds experienced breathing difficulties. (**Panel B**) shows the spleen of a CD4^+^/CD8^+^ T cell depleted bird at termination. Left: spleen from a CD4^+^ T cell depleted bird; right: spleen from CD4^+^/CD8^+^ T cell depleted bird. This contrasts with MDV-infected birds where the spleen is enlarged (splenomegaly), and the thymus and bursa are atrophied. (**Panel C**) depicts the bursa of a CD4^+^/CD8^+^ T cell depleted bird. Although the spleen tissues from these birds were negative for MDV genome, the bursas, like the spleens, were severely atrophied. Left: bursa from a CD4^+^ T cell depleted bird; right: bursa from a CD4^+^/CD8^+^ T cell depleted bird.

**Table 1 viruses-15-00648-t001:** T cell depletion, vaccination, challenge, and tissue sample collection.

Chicken Line 15I_5_ × 7_1_ Hatch day 26 Birds/Isolator	T Cell Depletion IP Day of Hatch & 3 dph *	T Cell Depletion IP/IV 6 & 9 dph	Flow to Verify T Cell Depletion 11 dph	Vaccination Rispens 2000 pfu 12 dph	T Cell Depletion IP/IV 12 & 15 dph	Flow to Verify T Cell Recovery 28 dph	Challenge rMd5 1000 pfu 29 dph	Samples Collection: 5, 10, 20, & 57 dpi
A: Control No Treatment			√			√		√
B: Birds with Intact T Cells				√			√	√
C: Birds with Depleted CD4^+^ T Cells	√	√	√	√	√	√	√	√
D: Birds with Depleted CD8^+^ T Cells	√	√	√	√	√	√	√	√
E: Birds with Depleted CD4^+^/CD8^+^ T Cells	√	√	√	√	√	√	√	√
F: Birds with Intact T Cells							√	√

* dph: days post hatch.

**Table 2 viruses-15-00648-t002:** Anti-chicken CD4 and CD8 treatment schedule.

Ab Treatment	Day of Hatch	3 dph *	6 dph	9 dph	11 dpi Flowcytometry to Verify T Cell Depletion	12 dph Vaccination CVI/Rispens 2000 pfu/bird	12 dph	15 dph	28 dph Flowcytometry to Verify the Recovery of T Cells	29 dph Challenge rMd5 1000 pfu/bird
Anti-CD4 Injection	200 μL ^a^ IP ^b^	200 μL IP	100 μL IV ^c^ 200 μL IP	100 μL IV 200 μL IP	√	√	100 μL IV 200 μL IP	100 μL IV 200 μL IP	√	√
Anti-CD8 Injection	200 μL IP	200 μL IP	100 μL IV 200 μL IP	100 μL IV 200 μL IP	√	√	100 μL IV 200 μL IP	100 μL IV 200 μL IP	√	√
Anti-CD4/CD8 Injection	200 μL anti-CD4 IP 200 μL anti-CD8 IP	200 μL anti-CD4 IP 200 μL anti-CD8 IP	100 μL anti-CD4 IV 100 μL anti-CD8 IV 200 μL anti-CD4 IP 200 μL anti-CD8 IP	100 μL anti-CD4 IV 100 μL anti-CD8 IV 200 μL anti-CD4 IP 200 μL anti-CD8 IP	√	√	100 μL anti-CD4 IV 100 μL anti-CD8 IV 200 μL anti-CD4 IP 200 μL anti-CD8 IP	100 μL anti-CD4 IV 100 μL anti-CD8 IV 200 μL anti-CD4 IP 200 μL anti-CD8 IP	√	√

* dph: days post hatch, ^a^ 1 mg/100 μL; ^b^ IP: intraperitoneal; ^c^ IV: intravenous.

**Table 3 viruses-15-00648-t003:** Viral genome copy number in the skin tissues of treated birds at 57 dpi.

Group *	Genome Copy Number at 57 dpi **
Control	0
Intact T Cell—Vaccinated, Challenged	78.44
CD4-Depleted—Vaccinated, Challenged	71.32
CD8-Depleted—Vaccinated, Challenged	0
CD4-CD8-Depleted—Vaccinated, Challenged	0
Intact T Cell—Challenged	1124.22

* 3 samples/virus group tested in triplicates; ** (MDV genome copy number/GAPDH copy number) × 1000.

**Table 4 viruses-15-00648-t004:** Protection efficacy of vaccination in T cell depleted and challenged birds.

Chicken Line 15I_5_ × 7_1_ (Ab-)	Antibody Treatment	Vaccination CVI988/Rispens (2000 pfu)	Challenge rMd5 (1000 pfu)	MD Incidence	Protection Efficacy
**Control Birds, Non-Treated**	None	None	None	N/A	N/A
**Birds with Intact T Cells, Vaccinated/Challenged**	None	√	√	None	100%
**Birds with CD4^+^ T Cells Depleted, Vaccinated/Challenged**	Anti-CD4	√	√	None	100%
**Birds with CD8^+^ T Cells Depleted, Vaccinated/Challenged**	Anti-CD8	√	√	None	100%
**Birds with CD4^+^/CD8^+^ T Cells Depleted, Vaccinated/Challenged**	Anti-CD4/CD8	√	√	None	??
**Intact T Cells Birds, Non-Vaccinated, Challenged**	None	None	√	100%	None

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
