# Peer review of "Role of T Cells in Vaccine-Mediated Immunity against Marek’s Disease"

_viruses, 2023, doi:10.3390/v15030648_

Round 1
Reviewer 1 Report (Previous Reviewer 1)
Most issues have been clarified and changed… I only have minor points that should be addressed before this article can be published in Viruses.
Line 308: “Purification and binding specificity of anti-CD4 and anti-CD-8 mAbs” should be subsection 3.1. – and the following sections should be renumbered accordingly. You could also combine the paragraph with “3.5. Flow cytometric analysis of binding specificity of anti-CD4 mAb”, line 384f.
Fig. 1E: The number of CD4 T cells in CD4-depleted birds is 0.02 (see previous version of the manuscript).
I suggest you should be consistent in capitalizing of plot- and axes labels (Fig. 1 (where you also left the paragraph marks and other hidden formatting symbols!!)… and Fig. 2). Please also include horizontal lines in Fig. 1E!
You still use “PBMN” in lines 340 and 369.
The line numbers are integrated into Tab. 3. That should be corrected.
An asterisk is missing before “3 samples/virus group tested in triplicates.”
Include SD in column “Genome copy number at 57 dpi**”
Lines 391-401 (now lines 494-504) are redundant and could be omitted. Your response was “Sorry, but we could not match these lines to a specific section of the discussion that we could delete or revise.”… I was referring to your introduction… but if you want to keep this paragraph: fine by me.
It has recently been shown that a protective immunity against Marek's disease is mediated by a subset of activated T cells – these findings should be briefly discussed…
It would be interesting to briefly discuss the benefits of using transgenic chickens to study the role of different immune cell subsets, as has been done with B-cell knockout chickens.
Author Response
Comments and Suggestions for Authors
Most issues have been clarified and changed… I only have minor points that should be addressed before this article can be published in Viruses.
Line 308: “Purification and binding specificity of anti-CD4 and anti-CD-8 mAbs” should be subsection 3.1. – and the following sections should be renumbered accordingly. You could also combine the paragraph with “3.5. Flow cytometric analysis of binding specificity of anti-CD4 mAb”, line 384f.
Authors’ response: Correction is made. Respectfully, we decided to keep the two sections separate.
Fig. 1E: The number of CD4 T cells in CD4-depleted birds is 0.02 (see previous version of the manuscript).
Authors’ response: The Figure 1 is replaced with corrected one. Thank you.
I suggest you should be consistent in capitalizing of plot- and axes labels (Fig. 1 (where you also left the paragraph marks and other hidden formatting symbols!!)… and Fig. 2). Please also include horizontal lines in Fig. 1E!
Authors’ response: Correction is made. Please see Figs. 1 and 2. Thank you.
You still use “PBMN” in lines 340 and 369.
Authors’ response: Correction is made. Thank you.
The line numbers are integrated into Tab. 3. That should be corrected.
An asterisk is missing before “3 samples/virus group tested in triplicates.”
Include SD in column “Genome copy number at 57 dpi**”
Authors’ response: Correction is made in Table 3 and Figure 5.
Lines 391-401 (now lines 494-504) are redundant and could be omitted. Your response was “Sorry, but we could not match these lines to a specific section of the discussion that we could delete or revise.”… I was referring to your introduction… but if you want to keep this paragraph: fine by me.
Authors response: Respectfully, we decided to keep the section as it emphasizes on the need for deciphering the underlying mechanism of vaccine protection and development of new recombinant vaccines.
It has recently been shown that a protective immunity against Marek's disease is mediated by a subset of activated T cells – these findings should be briefly discussed…
Authors’ Response: Additional material is added to the Discussion section. Please see lines 570-580.
It would be interesting to briefly discuss the benefits of using transgenic chickens to study the role of different immune cell subsets, as has been done with B-cell knockout chickens.
Authors’ Response: Additional material is added to the Discussion section. Please see lines 583-587.
Reviewer 2 Report (Previous Reviewer 2)
MDV, the etiological agent of MD, is a cell-associated α-herpesvirus that causes paralysis, anemia, immunosuppression, and T cell lymphomas in visceral organs of domestic chickens. Although vaccines have been used to control MD since the late 1960’s, the molecular mechanism of vaccine-induced protection is not well known. Vaccination prevents MDV-induced tumor formation, but not infection and transmission. Continuous evolution of MDV has been a major concern for the poultry industry.
I have read the revised manuscript and the authors’ point-by-point responses carefully. I found the authors had expanded and explained my questions in revised Introduction and Discussion sections. I believe the latest manuscript with the revisions has been significantly improved and now warrants publication in viruses.
Author Response
I have read the revised manuscript and the authors’ point-by-point responses carefully. I found the authors had expanded and explained my questions in revised Introduction and Discussion sections. I believe the latest manuscript with the revisions has been significantly improved and now warrants publication in viruses.
No comments/suggestion was raised by the Reviewer. Thank you.
This manuscript is a resubmission of an earlier submission. The following is a list of the peer review reports and author responses from that submission.
Round 1
Reviewer 1 Report
This manuscript by Heidari et al aims at deciphering the contribution of T cell subsets to vaccine protection against MD and MDV-induced tumors. They conclude that both CD4 and CD8 T cells do not play a role in vaccine protection upon CVI vaccination of a highly inbred, MDV-susceptible chicken line.
Due to insufficient CD8 T cell depletion, the authors cannot make that claim.
Moreover, I am listing various issues I have with this manuscript that should be taken care of (major and minor).
Line 28f: Recent data from the author’s lab (Avian Disease and Oncology Laboratory) showed something different… Umthong et al showed: "In addition, we provide evidence of the vital role that CD8+ T cells play in MD prevention with respect to anti-viral and anti-tumor responses in both unvaccinated and MD vaccinated birds." The authors should extensively discuss the differences between the two approaches and data interpretations.
Line 41f: The current ICTV nomenclature assigned the Mardiviruses into species and not serotypes – please revise.
The paragraph starting in line 54 is veeery similar to a paragraph in doi: 10.1016/j.jvacx.2021.100128. I suggest that it should be revised and some recent data on the role of NK cells and IFN-gamma during MDV infection could be included.
The authors could put their work in a historical perspective by integrating previously published papers that ask related questions:
J. M. Sharma, K. Nazerian, R. L. Witter, Reduced Incidence of Marek's Disease Gross Lymphomas in T-Cell-Depleted Chickens, JNCI: Journal of the National Cancer Institute, Volume 58, Issue 3, March 1977, Pages 689–692, https://doi.org/10.1093/jnci/58.3.689
B. W. Calnek, J. Fabricant, K. A. Schat, K. K. Murthy, Rejection of a Transplantable Marek's Disease Lymphoma in Normal Versus Immunologically Deficient Chickens, JNCI: Journal of the National Cancer Institute, Volume 60, Issue 3, March 1978, Pages 623–631, https://doi.org/10.1093/jnci/60.3.623
Morimura, T., Cho, KO., Kudo, Y. et al. Anti-viral and anti-tumor effects induced by an attenuated Marek’s disease virus in CD4- or CD8-deficient chickens. Arch. Virol. 144, 1809–1818 (1999). https://doi.org/10.1007/s007050050705
Line 64: is it only a threat to the US poultry industry? Some other countries have MDV issues too…
Line 93: In Tables 1 and 2, the authors state that the chicks were vaccinated on July 2 and challenged on July 19. That is not 10 days post-vaccination!
Line 106f: Umthong et al only provide details on purification and characterization for CD8 mAbs. The authors should provide all data for the CD4 mAbs (see Fig 2 Umthong et al.).
Line 114f: How where the routes of delivery and concentrations of mAbs needed for efficient depletion of CD4+ and CD8+ T cells optimized? Can the authors also show these data?
Line 123: Does PBMN stand for PBMC? Why did the authors choose this rare abbreviation and not PBMC?
Line 150: Can the authors please double-check the GAPDH reverse primer sequence because I could neither find it via google nor through NCBI BLAST or in the NCBI Gallus gallus ref seq (only in the GAPDH mRNA…).
Line 186 and Tables 1 and 2: To me, it looks like the last two treatments were on the day of vaccination and 3 days later. Please double-check and correct these numbers/statements.
Table 1: What does “Hatch: 6/20” stand for? A date? I would remove all the dates from this table and rather indicate the respective day post-hatching. Same accounts for Table 2.
Line 206: The statement “significant reduction” requires statistical analyses. Please perform appropriate stats or revise the sentence. Same accounts for lines 217, 221, 224, 298, 418, and 450.
Line 208: with “10 days post antibody treatment” the authors refer to 10 days after the first antibody treatment or 10 days post-hatching, correct? Please revise.
Line 211: Instead of “approximately two weeks”, the authors should write “13 days”.
Figures 1 and 2 are supposed to be representative FACS plots to Figure 3. So is Figure 4 to Figure 5. Yet, the percentages on the FACS plots are exactly the same as in the bar graphs. Importantly, the authors show SDs or SEMs in the bar graphs that originate from what exactly? Repeated measurements of the same sample? Because the authors pooled three birds per each time point, that is the most likely explanation. If they indeed are repeated measurements of the same sample the authors should provide an n in the figure legend and explain the remarkable differences between the measurements (except for the B cells, maybe…) and show the individual measurements as dots to allow a better interpretation of these data. Moreover, Umthong et al used exactly the same birds but with pretty different cell counts/percentages. The same holds true when comparing these numbers to the authors’ B cell paper in Vaccine X. How do the authors explain that?
Line 224f and Figure 3: is a roughly 2-3 fold reduction in overall CD8 T cell levels significant? Wouldn’t the authors have to test many more birds to make that statement?
Figure 3: none of these birds has been vaccinated and challenged yet. Please revise figure and legend.
Line 251: “terminated 3 days post-vaccination” would be correct, see Tables 1 and 2!
Paragraph starting in line 280: the authors could quantify that using qPCR (as in doi: 10.1371/journal.ppat.1010745, for example). The same should be done with the spleen DNA samples. The authors could nicely quantify the MDV genome levels and perform proper statistics!
Paragraph starting in line 307: The pp38 protein of the MDV serotype 1 vaccine strain CVI988/Rispens differs by one amino acid when compared to the pathogenic strains of MDV, see doi: 10.1007/s11262-005-2202-2. Did the authors consider that while designing the pp38 primers? That would be important to note!!!
Figure 9: wrong marker labeling.
Figure 10: no red arrows (see legend, line 352).
These Figures could be combined to improve their quality and professionalism:
Figures 1, 2 and 3
Figures 4 and 5
Figures 7 to 10
Figures 11 to 13
Line 366: Please compare these findings to other studies that performed antibody-mediated CD4/CD8 T cell depletions. Could this phenotype originate from the depletion? Or is it infection-associated? I suggest the authors test this in an additional experiment comparing infected vs non-infected vaccinated CD4/CD8 T cell-depleted birds.
Throughout the manuscript: the term absence is not used correctly, see lines 25, 360, or 415, for example. The authors showed by FACS that the chickens still have considerable amounts of T cells present – especially CD8 T cells!
Lines 391-401 are redundant and could be omitted.
The last part of the discussion merely is a repeat of the presentation of the results (lines 448 to 467). It would be nice if the authors rewrite that section and rather put their data into context.
Reviewer 2 Report
MDV, the etiological agent of MD, is a cell-associated α-herpesvirus that causes paralysis, anemia, immunosuppression, and T cell lymphomas in visceral organs of domestic chickens. Although vaccines have been used to control MD since the late 1960’s, the molecular mechanism of vaccine-induced protection is not well known. Vaccination prevents MDV-induced tumor formation, but not infection and transmission. Continuous evolution of MDV has been a major concern for the poultry industry.
1. There are individual differences in experimental animals. How to ensure that all T cells can be completely depletion of circulating T cells through IP/IV injection of anti-chicken CD4 and CD8 monoclonal antibodies?
2. The vaccinated birds with combined depletion of CD4+ and CD8+ T cells were severely emaciated, with atrophied spleens and bursas. How about chickens growing in the absence of CD4 or CD8 T cells? Please explain this phenomenon.
3. According to the conclusion of this paper, CD4+ and CD8+ T lymphocytes did not play a critical role in vaccine-mediated protection against MDV-induced tumor development. Which kind(s) of cells do you think that play a role in MDV vaccine immune protection.
4. Cellular immunity plays an important role in MDV vaccine immunization. Please fully discuss in the discussion section which kind(s) of cells that may play a role in MDV vaccine immune protection.